# Prevention of Hypothermia in the Aftermath of Natural Disasters in Areas at Risk of Avalanches, Earthquakes, Tsunamis and Floods

**DOI:** 10.3390/ijerph19031098

**Published:** 2022-01-19

**Authors:** Kazue Oshiro, Yuichiro Tanioka, Jürg Schweizer, Ken Zafren, Hermann Brugger, Peter Paal

**Affiliations:** 1Department of Cardiovascular Medicine and Mountain Medicine, Research, and Survey Division, Hokkaido Ohno Memorial Hospital, Sapporo 063-0052, Japan; 2Faculty of Science, Institute of Seismology and Volcanology, Hokkaido University, Sapporo 060-0810, Japan; tanioka@sci.hokudai.ac.jp; 3WSL Institute for Snow and Avalanche Research SLF, 7260 Davos, Switzerland; schweizer@slf.ch; 4Department of Emergency Medicine, Stanford University Medical Center, Stanford, CA 94304, USA; zafren@stanford.edu; 5Alaska Native Medical Center, Anchorage, AK 99508, USA; 6Institute of Mountain Emergency Medicine, Eurac Research, 39100 Bolzano, Italy; hermann.brugger@eurac.edu; 7Department of Anesthesiology and Intensive Care Medicine, St. John of God Hospital, Paracelsus Medical University, 5020 Salzburg, Austria; peter.paal@icloud.com

**Keywords:** hypothermia, accidental, avalanche, earthquake, flooding, mountain, ocean, prevention, rescue, tsunami

## Abstract

Throughout history, accidental hypothermia has accompanied natural disasters in cold, temperate, and even subtropical regions. We conducted a non-systematic review of the causes and means of preventing accidental hypothermia after natural disasters caused by avalanches, earthquakes, tsunamis, and floods. Before a disaster occurs, preventive measures are required, such as accurate disaster risk analysis for given areas, hazard mapping and warning, protecting existing structures within hazard zones to the greatest extent possible, building structures outside hazard zones, and organising rapid and effective rescue. After the event, post hoc analyses of failures, and implementation of corrective actions will reduce the risk of accidental hypothermia in future disasters.

## 1. Introduction


**‘Only human beings can recognize catastrophes, provided they survive them; nature recognizes no catastrophes.’**

**Max Frisch, Man in the Holocene. 1979.**


Natural disasters have plagued humankind since the beginning of our species. Since modern humans spread out of Africa 60,000–70,000 years ago, they had to protect against cold with shelter, clothing, and fire. Natural disasters can destroy all personal belongings and infrastructure and can deprive humans of the possibility of protecting themselves from harsh environments. One consequence may be death from accidental hypothermia. This article gives an overview of the history, epidemiology, mechanisms, pathophysiology, rescue, and treatment of accidental hypothermia caused by natural disasters such as avalanches, earthquakes, tsunamis, and floods. We provide an approach to the prevention of accidental hypothermia. We also discuss the management of multiple casualties, which is often necessary after natural disasters. A multi-casualty incident is an event in which the number of victims exceeds or substantially stretches the technical and medical resources of the local rescue system [1]. We conducted a non-systematic review of the causes and means of preventing accidental hypothermia after natural disasters from avalanches, earthquakes, tsunamis, and floods.

## 2. Hypothermia Caused by Avalanches

Avalanches are one of the greatest natural threats to humans living in mountain areas. Avalanches have repeatedly caused disasters claiming multiple victims [2]. An avalanche can cause hypothermia by direct contact of a buried victim with snow or by exposure to the cold environment if a victim remains unburied or is trapped in a destroyed structure or vehicle with inadequate shelter and delayed rescue.

### 2.1. History and Epidemiology

Avalanches have claimed many lives, especially in mountain warfare. In 216 BC, Hannibal lost up to 20,000 soldiers to avalanches and cold when he crossed the Alps [3]. During World War I, in the Italian Dolomites, the death toll from avalanches amounted to several thousand soldiers [4].

During severe winter storms, widespread avalanche cycles have claimed many lives among residents of mountain settlements. For instance, in 1951, during the most disastrous winter in the twentieth century in the European Alps, avalanches destroyed 900 buildings and killed 256 people [5]. In 1962, in Peru, an ice avalanche from the Nevados Huascaran killed about 4000 people in the city of Ranrahirca [6]. In 1979, in India, a series of avalanches in the Lahaul Valley killed 200 people [3]. In 1991, an avalanche hit several villages in Bingol, Turkey, killing at least 255 people [7]. In 1999, catastrophic avalanches claimed 38 lives in the Austrian villages of Galtür and Valzur [8]. In 2012 and 2015, a series of avalanches destroyed several villages and killed approximately 400 people in Afghanistan [9,10]. In 2017, an avalanche killed 29 people in a hotel near Farindola, Italy [11]. In 2014, an ice avalanche hit the Khumbu Icefall in Nepal on the climbing route of Mt. Everest, killing 16 people [12].

### 2.2. Mechanisms of Avalanche Burial

Over the last several decades in Europe, the number of avalanche fatalities on roads and in villages has steadily declined because of preventive measures. Currently, about 90% of the avalanches affecting humans in Europe or North America are triggered by recreationists [13,14,15]. The number of people venturing into avalanche terrain is unknown even in Europe and North America. Between 1983 and 2015, avalanches claimed over 5000 lives in Europe and North America, with an average annual toll of about 130 in Europe and 36 in North America [16]. Recreationists account for more than 90% of avalanche fatalities in the European Alps [17] and in North America [18,19,20].

Worldwide, outside North America and Europe, mortality statistics are only rough estimates. Avalanche fatalities are not systematically reported in the highest mountain regions, including the Andes, the Karakoram, and the Himalayas, where even densely populated valleys and villages are not protected from avalanches. Occasionally, avalanche disasters claim many lives. With global warming, the risk of heavy, often wet, snowfall events, and consequently severe avalanche cycles, may increase in some mountain ranges. Safety precautions may have to be re-evaluated and increased [21]. Mixed avalanches, with snow, ice, and rock, may occur more frequently, sometimes causing glacial lake outburst floods (GLOFs) or forming dams that block rivers. These cascading hazards can lead to flooding, with far-reaching effects downstream [22].

### 2.3. Prevention and Mitigation of Avalanche Burial

Avalanche disasters during World War I contributed to the founding of the first Commission for Snow and Avalanche Research in Switzerland in 1931. In 1942, the Institute for Snow and Avalanche Research was established in Davos, Switzerland. Three years later, the first civil avalanche warning service in the world became operational. Other warning services were established over time in several other European countries [23]. Especially after 1951, public funds were invested in engineering works, such as for supporting structures above villages, snow sheds, and dams, and for planting protective forests. These measures have resulted in the almost complete elimination of avalanche disasters in the European Alps.

In general, strategies for dealing with avalanche hazards include preventing the release of avalanches, influencing their paths, and avoiding hazardous areas [24]. Accidents can be prevented by controlling avalanches, by regulating human presence, and by placing structures outside avalanche paths. These measures require avalanche forecasting, public warnings, and hazard mapping for land use planning. These are relatively inexpensive means of mitigation. On the other hand, engineering works are expensive and require significant resources [25]. If avalanche hazards are evaluated daily, temporary road closures and evacuation of buildings can also be effective mitigation measures. The duration of these disruptions can be limited by intentionally triggering avalanches using explosives.

In the mountain areas where residents and recreationists are threatened by avalanches and where avalanche mitigation is still in its infancy, the first step should be to establish weather forecast and avalanche warning services. Weather forecasts, as well as current avalanche warnings, should be publicly available and widely distributed. In populated areas with a high risk of avalanches, emergency response and evacuation plans are necessary. Local authorities should be specifically trained in avalanche risk management [26].

For recreationists, who account for most avalanche victims in Europe and North America, training in avalanche risk assessment is critical. Proper trip planning, adequate rescue equipment, and safe travel habits complement avalanche prevention during winter backcountry activities in avalanche terrain [27].

### 2.4. Pathophysiology of Avalanche Burial with a Focus on Accidental Hypothermia

A detailed description of the pathophysiology of avalanche burial and discussion of the avalanche survival curve are beyond the scope of this article. They can be found elsewhere [28,29]. Although avalanche injuries may not be severe enough to cause death, the depressed level of consciousness caused by trauma may decrease the probability of survival by predisposing to asphyxiation, limiting shivering, and accelerating cooling. Relatively mild maritime climates with high snow density are associated with a significantly earlier decrease in survival compared to continental climates, with low snow density and cold temperatures [28]. The longest survival of a completely buried avalanche victim in the open was 44 h [30]. The longest survival in a structure was 37 days [31]. Survival longer than 35 min has been reported in up to 25% of completely buried avalanche victims who were able to breathe under snow. The term ‘triple H syndrome’, referring to the combined effects of hypoxia, hypercapnia, and hypothermia, was coined in 2003 [32]. Avalanche victims breathing into an air pocket in low-density avalanche debris or into a large or open air pocket may be able to survive for hours [33,34]. Accidental hypothermia has been reported as the sole cause of death in only about 1% of completely buried avalanche victims. Trauma can increase the risk of hypothermia. A higher injury severity score is associated with a higher incidence and severity of hypothermia [35,36]. Hypothermia should be suspected in an avalanche victim presenting with an open (not obstructed) airway at extrication after burial for >60 min [37,38]. Individual cooling rates during snow burial vary widely, from 0.1 °C/h [39] to 9 °C/h [40,41]. It usually takes at least 1 h after avalanche burial for a victim to reach a core temperature < 30 °C [38]. During rescue operations there is an additional risk of post-extrication cooling when avalanche victims are exposed to cold air on the snow surface after being buried in snow. Post-extrication cooling can be more rapid than cooling in the snow because of wind and lower ambient temperatures [41]. Moving air on the skin causes more rapid cooling than still air [42]. Cooling rates have not been reported for people buried in houses or vehicles. Anecdotal and statistical evidence suggests that survivors in structures tend to survive longer than buried subjects in open terrain, at least partly because they are not in constant, direct contact with snow (Figure 1) [31,43].

### 2.5. Rescue and Treatment

In disasters, burial by avalanches in buildings or vehicles differs significantly from burial in the open. Victims buried in buildings or vehicles usually lack personal protective avalanche equipment [45,46,47,48]. Searching with avalanche transceivers or RECCO^®^ is not feasible [49]. For the first several hours or days, the first responders are often local lay rescuers who have survived the avalanche. As soon as professional rescuers arrive at the scene, disaster medicine with triage can begin [1]. The International Commission for Mountain Emergency Medicine (ICAR MedCom) has published guidelines for rescue and medical care in multi-casualty avalanche accidents [1]. The number of casualties defining a multi-casualty avalanche may be lower than in an urban environment because of limited resources. The principles of care for multi-casualty incidents (MCIs) in mountain areas are given in Table 1.

An avalanche accident should prompt a helicopter rescue, if possible. Helicopters can decrease the response times of and risks to the rescue teams by transporting rescuers safely above potentially hazardous terrain. ‘Safety first’ should be the guiding principle for rescue operations. The risks to rescue teams must be weighed against the potential benefits for the victims. Avalanche dogs and probing may be valuable tools for timely location of buried subjects [50]. Technical rescue may require heavy machinery and equipment. Special precautions may be required when accessing vehicles or entering buried structures. For a mass avalanche incident with insufficient medical resources, a rescue strategy was developed using the Monte Carlo method [51].

Once found, the treatment of victims buried in buildings or vehicles is similar to that of victims in open terrain. Unfortunately, no data have been published on methods of accessing victims buried in buildings or in vehicles. An extrication time of 7 min for the first vertical meter of snow burial has been reported in an experimental study, but this is not likely to apply in a real-world situation with hard avalanche debris [52]. The more deeply a victim is buried, the longer the extrication time, because a larger volume of snow must be removed.

After extrication, a victim should be insulated from cold using available materials, such as parkas, wind shells, hats, gloves, bivouac sacks, and sleeping bags, and with vapour barriers such as aluminium blankets. Medical treatment is beyond the scope of this article and is discussed in detail elsewhere [16,38,53].

### 2.6. Outcome

In avalanche disasters affecting structures and vehicles, the rate of long-term survival may be higher than in avalanches in open areas, partly because air pockets in structures are likely to be larger than air pockets in the open (Figure 1) [31,43]. The extrication times may be substantially longer in avalanches affecting structures and vehicles.

## 3. Hypothermia Caused by Earthquakes and Tsunamis

### 3.1. Key Challenges of Large Earthquakes

Earthquakes with associated tsunamis have killed more people than all other disasters combined. Millions of earthquakes occur every year [54]. Most are small and cause no damage. Over 80% of the fatalities from earthquakes have been in China, Japan, Pakistan, Turkey, countries of the former USSR, Peru, Chile, and Italy. Earthquakes and tsunamis are rapid-onset disasters. From 1994 to 2013, earthquakes were responsible for an estimated 1.35 million deaths and displacement of an estimated 218 million people [55]. In addition to tsunamis, earthquakes can cause snow or ice avalanches, as well as GLOFs. All these disasters have the potential to cause accidental hypothermia in the people who initially survive an earthquake. Most people who suffer from hypothermia in earthquakes associated with tsunamis become hypothermic by being caught in the tsunami. Data regarding hypothermia caused by earthquakes and associated disasters are very limited.

#### 3.1.1. Irpinia Earthquake 1980

One report of accidental hypothermia after an earthquake concerned the 1980 Irpinia earthquake in southern Italy [56]. On 23 November 1980, this magnitude 6.9 earthquake killed almost 2500 people, injured at least 7700, and left 250,000 people homeless. Nine children were buried for 7 to 27 h before being rescued alive. All were wearing light pyjamas or flannels. The lowest temperature overnight was 7.7 °C. Seven of the nine children were buried and immobilised for longer than 12 h and were mildly hypothermic on arrival to the hospital. All were rewarmed without incident using radiant infant warmers.

#### 3.1.2. Kashmir Earthquake 2005

The magnitude 9.0 South Asian earthquake struck Kashmir on 8 October 2005. The official death toll was over 73,000 in the part of Kashmir administered by Pakistan and almost 1400 in the part administered by India [57]. An estimated 3.3 million people were left homeless. Three months after the quake, most of the people left homeless were still living in tents. Night-time temperatures were below freezing. In mountain villages, temperatures fell to as low as −13 °C. Few of the tents were winterised or designed for arctic-type cold. Relief efforts were thwarted by landslides caused by heavy snow and rain. Supplies and personnel could not be airlifted because of fog. Although there was enough food, there was a shortage of warm clothing and blankets. Many tents collapsed under snow. A news report on 28 November 2005 estimated that more than 100 people had been brought to hospitals with hypothermia and respiratory diseases.

#### 3.1.3. Great East Japan Earthquake 2011

Probably the best studied tsunami followed the Great East Japan earthquake on 11 March 2011. There were about 16,000 fatalities. Over 90% were caused by drowning [58]. The ratio of tsunami-related deaths to earthquake-related deaths was 9:1. The ratio of injuries to deaths was 1:3.9. If missing persons are included, the ratio was 1:3.3. This contrasts with a ratio of 1:0.15 injuries to deaths after the Great Hanshin earthquake of 1995 [59]. The occurrence of a tsunami increased the number of fatalities. Causes of death were not investigated in any detail. Autopsies were not conducted. As with most natural disasters, corpses were examined primarily to identify the victims. Interviews with medical examiners suggested that diagnoses of ‘death by drowning’ may have included cases of death by hypothermia and cases in which victims developed hypothermia before drowning [60]. Hypothermia accounted for 0.2% of deaths reported in the three hardest hit prefectures [61].

Victims of hypothermia caused by earthquakes and tsunamis can be broadly divided into those who developed hypothermia after gradual exposure to cold, as a result of their homes collapsing or damage to infrastructure, and those who were caught in the tsunami. The average temperature during the 24 h after the earthquake was 0.8 °C. The lowest temperature was −1.3 °C. The average wind speed was 5.8 m/s, with a maximum wind speed of 9.7 m/s. During early March, the sea surface temperature is typically between 5.4 °C and 8.3 °C [62]. The likelihood that hypothermia will develop increases as ambient temperature decreases and is increased by convective heat loss caused by wind and evaporative heat loss from wet clothing [63]. The risk of hypothermia increases during immersion in water colder than 18 °C [64,65]. According to a survey of 134 patients in the hardest hit prefecture who received medical care at a hospital within 72 h after the earthquake, hypothermia caused by the tsunami accounted for about three-fourths (45 of 59) of all admissions. Most patients arrived at the hospital within 24 h [66]. The number of hypothermia cases occurring indoors increased with time, especially in victims with underlying conditions and in victims requiring assistance for activities of daily living who were living in their own homes or in shelters after evacuating their homes [66]. Seventy-seven of 91 (85%) patients were cold stressed at presentation (core temperature 35–35.9 °C—32 of 91) or mildly hypothermic (core temperature 32–35 °C—45 of 91) [66]. The predominance of patients with cold stress or mild hypothermia likely reflects the limited number of victims taken to hospitals during the early phase of the disaster. Land approaches were blocked when roads collapsed and were submerged. Air rescue was the primary means of reaching survivors. Air rescue was limited during the night. In a large-scale disaster, there is very little chance that hypothermic victims without vital signs will be resuscitated or transferred to a hospital [30,38]. When it is unclear if the cause of cardiac arrest is drowning or hypothermia, it is difficult to identify victims who might be candidates for extracorporeal life support (ECLS) rewarming [67]. Even if hypothermic victims are transferred, ECLS treatment may be impossible because of power outages. In the Great East Japan earthquake, about 80% of hospitals and 30% of clinics were damaged. Almost all medical facilities in coastal areas had limited capabilities. Water and power outages, including interruptions of natural gas pipelines, were widespread. It took several days before most medical facilities were able to operate normally.

Only 4 of 91 (4%) hypothermic patients arriving at hospital within 72 h of the earthquake died. The low mortality likely reflects the high proportion of cold stressed and mildly hypothermic patients. Patients without vital signs were not transferred to hospitals and were not represented in this in-hospital study [66]. Hospital treatment is limited when infrastructure is compromised because of limited rewarming capabilities, problems in identifying patients with underlying conditions and injuries, and the difficulties of allocating available resources. Most medical facilities also had little experience treating hypothermic patients.

The situation is different during a small-scale disaster with few casualties, in which the response capabilities of the area are not overwhelmed. A wider range of victims can access medical resources. Victims in hypothermic cardiac arrest might receive medical treatment with ECLS rewarming. The allocation of ECLS rewarming should be guided by outcome stratification using a validated tool, such as the Hypothermia Outcome Prediction after Extracorporeal Life Support (HOPE) score [68].

## 4. Hypothermia Caused by Flooding

Worldwide, floods are the most common disasters. Climate change increases the risk of floods by increasing precipitation [69]. Areas at special risk are low-lying areas near oceans or below sea level, such as parts of Bangladesh and the Netherlands, along rivers, and in or below mountain ranges in places where water can accumulate. The Netherlands is one of the countries with the highest risk of floods, because almost one-third of the country is below sea level. The Netherlands are protected by a sophisticated system of dams and water management [70].

In Europe, floods are the most common natural disasters. The most severe European flood of the last 100 years was the 1953 North Sea flood, caused by a combination of wind, high tides, and low atmospheric pressure. It flooded land up to 5.6 m above mean sea level. In the Netherlands, the 1953 North Sea flood claimed 2551 lives, flooded 9% of the Dutch farmland, drowned 30,000 animals, damaged 47,300 structures, and destroyed 10,000 buildings. There were 28 fatalities in Belgium, 307 in England, and 19 in Scotland [71,72,73]. Another 230 people died on boats. In the aftermath, England and the Netherlands reinforced their coastal defences with storm surge barriers. England built the Thames and Humber Estuaries. The Netherlands developed the Delta Works. In Italy, 68% of municipalities are at high risk of hydrological disasters. In 2021, heavy rainfall in Belgium, Germany, and the Netherlands again caused severe destruction and disruption.

North American and Asian countries have been struck by floods, often when dams were breached after hurricanes or typhoons. Hurricane Katrina claimed 1464 lives and destroyed vast low-lying sections of New Orleans [74]. The immediate health impacts of floods include drowning and other injuries, heart attacks, and accidental hypothermia. Indirect and delayed effects include displacement, starvation, unemployment, mental health problems (including post-traumatic stress disorder), respiratory diseases, allergic reactions, and water-borne infectious diseases [69].

In the US, between 2006 and 2010, 2000 US residents died because of weather events. Six percent of deaths were attributed to floods, storms, and lightning [75,76]. In the present climate crisis, rising seawater levels and a higher incidence of extreme precipitation are increasing the likelihood of floods along coastlines and riverbeds [77].

## 5. Treatment of Hypothermia after a Natural Disaster

Treatment strategies should be established with the assumption that infrastructure will be partly or completely disrupted over a large area. Precautions should be taken so that as few victims as possible become cold stressed or hypothermic. In the Great East Japan earthquake, 500,000 blankets were distributed [78]. Patients with cold stress or mild hypothermia were passively rewarmed with blankets in hospitals. Hospital treatment should be limited initially to hypothermic patients with underlying conditions or associated injuries and to victims with moderate or severe hypothermia. Medical professionals experienced in treating hypothermia should be deployed to evacuation shelters to help victims with mild symptoms self-treat and to identify victims requiring care at medical facilities. Wet clothing should be removed as soon as possible. If a change of clothing is not available [30,38], victims should be insulated with blankets [30]. Heating systems will likely not be available at evacuation shelters. Without electricity, forced air warming devices will not function. Chemical or battery-powered heat packs that do not require an external source of electricity or hot water bottles should be stockpiled for active external rewarming. A review of the diagnosis and treatment of accidental hypothermia can be found in this special issue [79].

## 6. Prevention and Mitigation

### 6.1. Earthquakes

There is usually little or no warning before an earthquake. Earthquakes, tsunamis, and floods that occur at night when people are indoors and sleeping usually cause more injuries and deaths than disasters that occur during the day. The Great Earthquake in Nepal on 25 April 2015 took place at about noon on a Saturday, the one day of the week when schools and businesses are closed in Nepal. Children were not in school. Most people in rural areas were outdoors, working in the fields. About 8000 people died. The toll would have been far higher if the earthquake had occurred on a normal working day or at night. Earthquakes also tend to have a higher mortality in winter, because more people are indoors during the day.

Earthquake early warning systems typically give less than a minute notice of an impending earthquake, but have saved many lives. Education about safety during an earthquake and about avoiding tsunamis is critical to preventing injuries and deaths (Table 2).

Damage to buildings is a major cause of injuries and deaths. Adequate engineering of buildings, roads, and embankments can prevent injuries and deaths. Needs in humanitarian crises, such as earthquakes and tsunamis, include shelter and security, food and water, and sanitation. To prevent hypothermia, adequate shelter should be prepared in advance. Food and essential non-food items should be stockpiled. Essential non-food items include clothing, bedding, cookware, plates, utensils, and soap. Sanitation is important to prevent diseases that could contribute to hypothermia as well as morbidity or mortality.

### 6.2. Tsunamis

To prevent hypothermia caused by tsunamis, susceptible areas should be identified. Tsunamis are rare events, occurring about once in a hundred years at any given location. Areas along the Pacific coasts of Canada, Alaska, Russia, and northern Japan, and along the coast of the Japan Sea, including areas of Japan, Russia, and Korea, have conditions in which hypothermia could occur after tsunamis (Figure 2).

In the cold areas that are vulnerable to large tsunamis, temporary shelters should be established on elevated land and should be supplied with rewarming equipment and adequate food. It is difficult to raise the temperature inside a large evacuation shelter such as a gymnasium. Evacuees, especially children and people with underlying conditions, may not be able to tolerate long stays. Evacuees should only stay in evacuation shelters for a short period of time before being transferred to accommodations with better infrastructure, such as hotels in unaffected areas. People who are not independent and who are living in their own homes or in nursing facilities are likely to remain where they live, unless an earthquake makes these structures uninhabitable. Food and water should be stockpiled, as should equipment to heat surviving structures if electricity and gas lines are disrupted. To prepare for temporary disruptions to communications, systems, such as global-positioning-system-enabled mobile phones, should be put in place to help victims summon help.

### 6.3. Floods

Early warning systems should be established for heavy rainfall and storms. Warning systems and coastline protections have been installed in many developed countries, and, to a lesser extent in developing countries, such as Bangladesh. Unfortunately, many early warning and coastline protections are rudimentary, requiring improvements to save more lives. Devastating floods in July 2021 along the Ahr and adjacent rivers in western Germany and eastern Belgium claimed more than 200 lives [80]. The floods caused over 7 billion euros in damages. The Ahr River had previously had a severe flood in 1910 that killed 200 people. In the future, housing and critical infrastructure should be built on high ground well above past flood levels.

## 7. Conclusions

Throughout history, accidental hypothermia has accompanied natural disasters in cold, temperate, and even subtropical regions. We have explored the causes and means of preventing accidental hypothermia after natural disasters caused by avalanches, earthquakes, tsunamis, and floods. Before a disaster occurs, preventive measures are required, such as accurate disaster risk analysis for given areas, hazard mapping and warning, protecting existing structures within hazard zones to the greatest extent possible, building structures outside hazard zones, and organising rapid and effective rescue. After the event, post hoc analyses of failures and implementation of corrective actions will reduce the risk of accidental hypothermia in future disasters.

## Figures and Tables

**Figure 1 ijerph-19-01098-f001:**
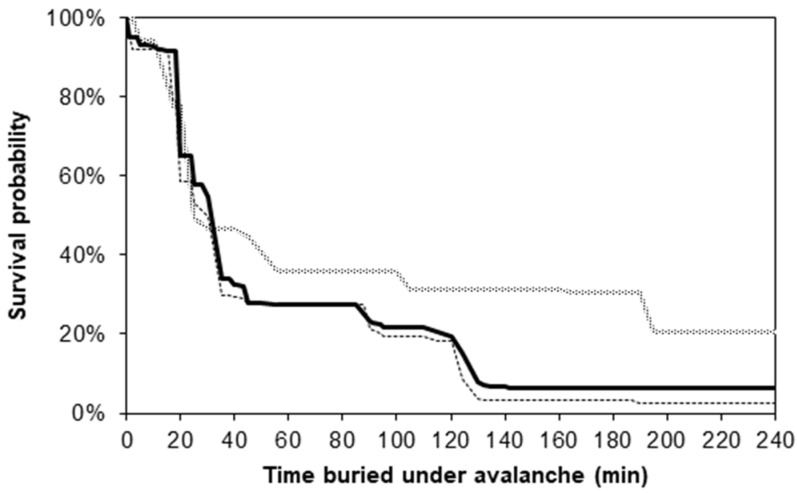
Survival probability for avalanche victims completely buried under the snow in Switzerland from 1981 to 1998 (*n* = 735) in relation to time (minutes), comparing victims buried in open areas (black curve, *n* = 638) with victims buried in buildings or vehicles (grey curve, *n* = 97). Median extrication times were 37 min in open areas and 56 min from buildings or vehicles (*p* = 0.17, Mann–Whitney U-test). In open areas, only 17% of all survivors were extricated after 35 min of burial, compared with 33% in buildings and on roads (*p* = 0.008; Pearson’s chi-square). The dotted curve represents the survival function for completely buried avalanche victims in open areas (*n* = 422) based on the Swiss data for 1981 to 1991, Reprinted with permission form ref. [43,44]. Copyright 2022 Copyright Clearance Center.

**Figure 2 ijerph-19-01098-f002:**
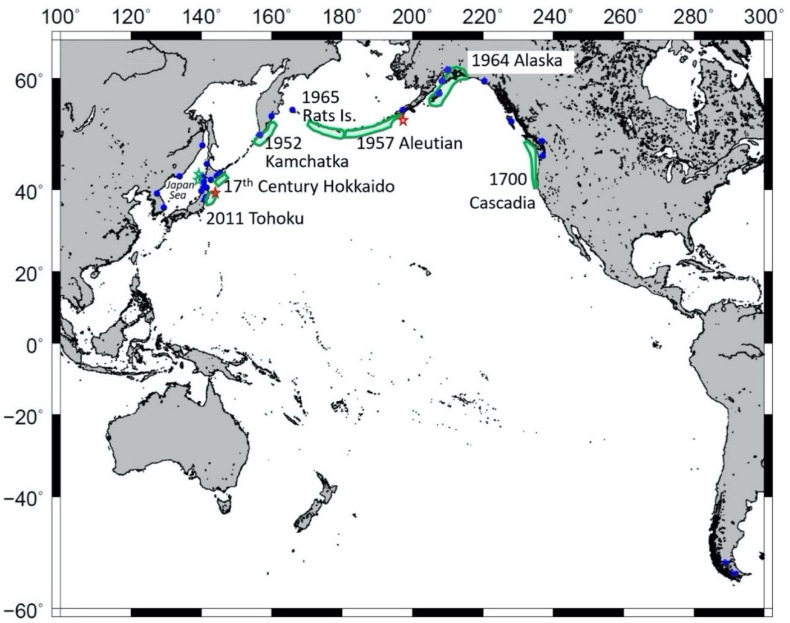
Areas in which victims may be vulnerable to hypothermia after tsunamis. Blue circles are locations in which victims are vulnerable to hypothermia after a tsunami, where monthly average minimum temperatures in 2020 were <0 °C in at least one month. Green shaded areas are source areas of historical earthquakes that caused tsunamis with maximum wave heights > 10 m. Green stars are epicentres of the 1983 Japan Sea and the 1993 Hokkaido Nansei-oki earthquakes (maximum wave heights > 10 m). Red stars are epicentres of the 1896 Sanriku and the 1946 Aleutian earthquakes with tsunamis (maximum wave heights > 30 m).

**Table 1 ijerph-19-01098-t001:** Principles and recommendations for the management of a multi-casualty incident (MCI) in a mountain area, Reprinted with permission form ref. [1]. Copyright 2022 Copyright Clearance Center.

General Principles and Recommendations in MCI Management
Declare an MCI. An MCI should be identified and the appropriate rescue organisations and hospitals should be alerted as soon as possible. Assess safety. Safety of the rescuers is the highest priority. Rescuers should not access an area if the risk to themselves is considered to be too high. Initial response. The initial response should focus on setting up a command-and-control structure, triage, and rapid life- or limb-saving interventions. Leadership and command. The medical commander should be trained in disaster medicine and in mountain rescue. The medical commander and leaders of the rescue services should all be at the same location on site to optimise cooperation. All should be easily identifiable. Ensuring effective communications. An effective communication structure should be implemented to support command and control. Triage. Effective triage tools adapted to mountain injuries should be implemented. Evacuations. Casualties should be evacuated to a safe area, then transferred to medical facilities appropriate to their medical needs. Identification and tracking. Tools that enable clear identification and tracking of casualties should be available for MCIs in mountains. Learning from experience. MCIs in mountain areas should be analysed afterwards and necessary improvements in prevention and management should be identified and implemented. Planning and training. Standard operating procedures should be available, familiar, and implemented with regular training exercises involving emergency services.
Specific principles and recommendations in MCI management in mountain areas. Environmental influence. Objective hazards that could affect rescuers and victims, should be constantly monitored and mitigated to the greatest possible extent. Use of helicopters. Helicopters with appropriate mountain rescue capabilities are often useful in MCIs. Coordination of helicopter and ground operations is critical. Communication devices and networks are helpful and should be widely used. Management of uninjured people. In a mountain environment, uninjured survivors should be considered as victims at risk. Psychological trauma is often present. Affected survivors should be treated, as necessary.
Specific MCIs in mountain areas. Avalanches. For a burial time <60 min, extrication is the first priority. Medical care should focus on victims with signs of life until enough resources are available to treat additional victims in cardiac arrest. For burial time >60 min, cardiopulmonary resuscitation (CPR) should only be started if the airway is patent. The use of a checklist may improve triage and treatment.

**Table 2 ijerph-19-01098-t002:** Strategies to prevent and mitigate hypothermia after natural disasters.

Areas Susceptible to Tsunamis Should Be Identified.
Future housing and critical infrastructure should be built in safe places. Existing structures in areas at risk should be moved or protected.
Early warning and protection systems should cover areas at risk.
Adequate engineering of buildings, roads, and embankments is required.
Education about safety and how to avoid natural disasters should be implemented for citizens in areas at risk.
Temporary shelters and sanitation should be established on elevated land with adequate food and water and should be supplied with rewarming equipment.
Essential non-food items, e.g., clothing and blankets, should be stockpiled.
Global positioning system (GPS)-capable communication systems should be available.

## Data Availability

Not applicable.

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
