# Peer review of "Prevention of Hypothermia in the Aftermath of Natural Disasters in Areas at Risk of Avalanches, Earthquakes, Tsunamis and Floods"

_ijerph, 2022, doi:10.3390/ijerph19031098_

Round 1
Reviewer 1 Report
1) Check for typos, e.g.:
-Title: “Hypothermia” should read “hypothermia”, “FloodS” should read “floods”, “tsunamiS” should read “tsunamis”
-Author names: “Yuuichiro” should read “Yuichiro”
-Abstract: “hypotherrmia” should read “hypothermia” (also in 4. Conclusions, which is identical to the Abstract)
There are many more trouthout the text, including double blank spaces (e.g. lines 39 and 100) or in line 55: there should be a blank space between “Alps!” and “[3]” and the full stop should instead be placed between “]“ and “During”.
2) Abstract, line 23: floods are missing.
3) 2.1: the first three sentences are about people killed by avalanches in wars but the subsequent sentences suddenly deal with incidences not related to wars. This section should be rewritten.
4) Line 74: write either “about” or “more than” but not both.
5) Lines 240–242: please add a reference and explain, why the risk of mechanical trauma may be higher in avalanches affecting structures and vehicles.
6) 3. Hypothermia Caused by Earthquakes, Tsunamis, and Floods:
In my opinion, it makes sense to describe floods should be described in a separate section as these – in contrast to tsunamis – are not associated with earthquakes.
7) 4. Conclusions: this section is identical to the Abstract, which does not really make sense in my opinion. Therefore, one of these sections should be rewritten.
Author Response
Reply to Reviewer 1
Thank you very much for the very valuable reviews. We revised the manuscript accordingly. Our replies to the reviewer below are highlighted in italics, all amendments in the manuscript are marked with the ‘track changes’ option or green in this document.
1) Check for typos, e.g.:
-Title: “Hypothermia” should read “hypothermia”, “FloodS” should read “floods”, “tsunamiS” should read “tsunamis”
Thank you for this hint, in the original .word file the words were written correctly. We checked the manuscript carefully for any mistakes.
Done, thank you.
-Author names: “Yuuichiro” should read “Yuichiro”
Thank you, done.
-Abstract: “hypotherrmia” should read “hypothermia” (also in 4. Conclusions, which is identical to the Abstract)
Done, thank you.
There are many more trouthout the text, including double blank spaces (e.g. lines 39 and 100) or in line 55: there should be a blank space between “Alps!” and “[3]” and the full stop should instead be placed between “]“ and “During”.
As a function of editing with double justified text. It is almost impossible to catch all double blank spaces. We checked the manuscript carefully again, (viewing it at 250%) thank you. We will rely on the copy editor to find any double blank spaces we missed. We have proofed the manuscript again to catch as many remaining typos as possible.
2) Abstract, line 23: floods are missing.
The word ‘floods’ has been added
3) 2.1: the first three sentences are about people killed by avalanches in wars but the subsequent sentences suddenly deal with incidences not related to wars. This section should be rewritten.
Thank you for this important hint. We have explained our thought process better and added the following:
2.1. History and epidemiology
Avalanches have claimed many lives, notably in mountain warfare. In 216 BC, Hannibal lost up to 20,000 soldiers because of avalanches and cold when he crossed the Alps [3]. During World War I, in the Italian Dolomites, the death toll from avalanches amounted to several thousand soldiers [4].
During severe winter storms, widespread avalanche cycles have claimed many lives among residents of mountain settlements. In 1951, during the most disastrous winter in the twentieth century in the European Alps, avalanches destroyed 900 buildings and killed 256 people [5].
4) Line 74: write either “about” or “more than” but not both.
Thank you, we deleted ‘about’
5) Lines 240–242: please add a reference and explain, why the risk of mechanical trauma may be higher in avalanches affecting structures and vehicles.
This assumption comes from the initial drop of the survival curve in 2001 immediately after burial in victims buried in vehicles or buildings (Fig.1) but this is not supported by in-hospital data about frequency and severity of trauma. We have deleted this sentence.
6) 3. Hypothermia Caused by Earthquakes, Tsunamis, and Floods:
In my opinion, it makes sense to describe floods should be described in a separate section as these – in contrast to tsunamis – are not associated with earthquakes.
Thank you for this important suggestion. We have added a separate section on floods, now ‘4. Hypothermia caused by flooding’, and within section ‘6. Prevention and mitigation’ section: ‘6.3 Floods’. Because of these new major rearrangements, the following sections were renumbered: ‘5. Treatment’, ‘6. Prevention and mitigation’, and ‘7. Conclusions’.
7) 4. Conclusions: this section is identical to the Abstract, which does not really make sense in my opinion. Therefore, one of these sections should be rewritten.
We intentionally came to the same conclusions in the main text and in the abstract. The conclusions should be identical, and this is also recommended when writing an article. Most of the readers will only read the abstract, and most often only the conclusions in the abstract. We would like to convey the proper conclusions of this work.
Reviewer 2 Report
This review by Oshiro, K et al., tackled the topic of prevention of hypothermia caused by aftermath of natural disasters such as avalanches, earthquakes, floods, and tsunamis. The review is written clearly, with summaries of related historical disasters and discussions of post disaster managements, focused on hypothermic conditions that contributed to the overall damages of the specific natural disaster. This is a good review in that the authors included up-to-date information regarding a specific event, meanwhile, also provided their own thoughts and discussed the possible better outcome if certain preventions would be included for the future disaster management.
There are a few minor/editorial mistakes that should be fixed:
- The title, should include the “aftermath” of natural disaster to make point clearer
- Also in title, “Floods” and “Tsunamis”, the capital “S” at the end of the words should be small “s”
- Line 39, there are more spaces than needed between “from” and “accidental”
- Line 42, there should a period and spaces between “tsunamis” and “We”
- Figure 1 legend, line 155, there are more spaces than needed between “survival” and “function”
- Figure 1, please add short legend around each curve for easy reading, although the details are included in the legend
- Table 1 title, line 173, please add (MCI) after multi-casualty incident to define “MCI” before using MCI later in the text
- Line 268, again there are more spaces than needed between “aftermath,” and “England”
- Line 21 and line 423, no coma should be added between “cold temperature”
Author Response
ijerph-1520060 - Prevention of Hypothermia in the Aftermath of Natural Disasters in Areas at risk of Avalanches, Earthquakes, Tsunamis and Floods
Reply to reviewer 2
Thank you very much for the very valuable reviews. We revised the manuscript accordingly. Our replies to the reviewers below are highlighted in italics, all amendments in the manuscript are marked with the ‘track changes’ option in this document.
This review by Oshiro, K et al., tackled the topic of prevention of hypothermia caused by aftermath of natural disasters such as avalanches, earthquakes, floods, and tsunamis. The review is written clearly, with summaries of related historical disasters and discussions of post disaster managements, focused on hypothermic conditions that contributed to the overall damages of the specific natural disaster. This is a good review in that the authors included up-to-date information regarding a specific event, meanwhile, also provided their own thoughts and discussed the possible better outcome if certain preventions would be included for the future disaster management.
Thank you very much for your positive comments.
There are a few minor/editorial mistakes that should be fixed:
- The title, should include the “aftermath” of natural disaster to make point clearer
This has been done, the new title now reads ‘Prevention of Hypothermia IN THE AFTERMATH OF natural disasters in areas at risk of avalanches, earthquakes, tsunamiS, AND FLOODS.’
- Also in title, “Floods” and “Tsunamis”, the capital “S” at the end of the words should be small “s”
Thank you for this observation, in the original .word file the words were written correctly. We checked the manuscript carefully. We have corrected this in the proof.
- Line 39, there are more spaces than needed between “from” and “accidental”
Revised, thank you
- Line 42, there should a period and spaces between “tsunamis” and “We”
Done
- Figure 1 legend, line 155, there are more spaces than needed between “survival” and “function”
Done
- Figure 1, please add short legend around each curve for easy reading, although the details are included in the legend
We did not change the original figure. The details are well described in the legend. Legends around the curves in the graph would be redundant.
- Table 1 title, line 173, please add (MCI) after multi-casualty incident to define “MCI” before using MCI later in the text
Done
- Line 268, again there are more spaces than needed between “aftermath,” and “England”
Revised, thank you
- Line 21 and line 423, no comma should be added between “cold temperature”
Done, thank you very much
Reviewer 3 Report
Dear authors,
Thank you for allowing me to revise this fascinating manuscript providing a summary of hypothermia due to environmental causes worldwide. Although the review of existing literature seems of outstanding quality, I have some concerns about the applicability of such a paper in clinical practice. The only reference to a post-event evaluation is present in the conclusions, but I would find some implications on how actions should be implemented to reduce the risk of death due to hypothermia. Moreover, I would know more about what kind of post-hoc analyses you suggested, and what kind of corrective actions should be implemented. I missed some methodological underpinning of this manuscript. I would like to know how you have performed the search and screened your results if it is a review. If it is a discussion paper, I would like some practical clinical tips to be used and to drive future research on this topic.
I would report some typos I have found along with the text.
Title: why have you used capital letters for Hypothermia, FloodS, and tsunamiS?
Abstract: Line 21, please remove the double r from hypothermia.
Page 3, Line 108 "is should be to" please correct.
Page 4-5, Table 1. Please, provide an explanation of the MCI acronym before using it extensively. Why is "psychological trauma" written in italics? Moreover, if these indications come from guidelines, why some of them don't give recommendations? e.g., assessing safety, use of helicopters, psychological trauma.
Page 5, line 209. Is this a new chapter? Maybe you should start with 2.6?
Page 6, line 235. Is "barrers" intended as "barriers"?
Page 6, line 267. Please, correct with "England".
Page 6, line 270. Please, put a comma after Netherlands.
Page 9, line 385. Please, correct with "higher".
I hope my comments will help you improve the quality of this manuscript.
Author Response
ijerph-1520060 - Prevention of Hypothermia in the Aftermath of Natural Disasters in Areas at risk of Avalanches, Earthquakes, Tsunamis and Floods
Reply to reviewer 3
Thank you very much for the very valuable reviews. We revised the manuscript accordingly. Our replies to the reviewers below are highlighted in italics, all amendments in the manuscript are marked with the ‘track changes’ option in this document.
Dear authors,
Thank you for allowing me to revise this fascinating manuscript providing a summary of hypothermia due to environmental causes worldwide. Although the review of existing literature seems of outstanding quality, I have some concerns about the applicability of such a paper in clinical practice. The only reference to a post-event evaluation is present in the conclusions, but I would find some implications on how actions should be implemented to reduce the risk of death due to hypothermia.
We have expanded the prevention and mitigation section. To our knowledge there are no peer-reviewed articles on this topic, which could be cited. Actions to be taken are now highlighted in the additional table 2.
Moreover, I would know more about what kind of post-hoc analyses you suggested, and what kind of corrective actions should be implemented.
Thank you for this important suggestion, we expanded this topic in the ‘6. Prevention and mitigation’ section. Table 2 has been added, which summarizes strategies to prevent and mitigate hypothermia after natural disasters.
I missed some methodological underpinning of this manuscript. I would like to know how you have performed the search and screened your results if it is a review.
This is a non-systematic review. There is almost no medical peer reviewed literature on this topic. We felt the urgency to present for the first time this topic to a wider audience to stimulate awareness and research. We have added this information to the abstract and the introduction.
If it is a discussion paper, I would like some practical clinical tips to be used and to drive future research on this topic.
This is not a discussion but a non-systematic review. Thank you.
I would report some typos I have found along with the text.
Thank you.
Title: why have you used capital letters for Hypothermia, FloodS, and tsunamiS?
Thank you for this observation. in the original .word file the words were written correctly. We checked and corrected the proof.
Abstract: Line 21, please remove the double r from hypothermia.
Done.
Page 3, Line 108 "is should be to" please correct.
Done, thank you.
Page 4-5, Table 1. Please, provide an explanation of the MCI acronym before using it extensively.
Done.
Why is "psychological trauma" written in italics?
Italics was a mistake, the font has been set to non-italics, thank you.
Moreover, if these indications come from guidelines, why some of them don't give recommendations? e.g., assessing safety, use of helicopters, psychological trauma.
Thank you for this important suggestion. We have revised the table to better reflect principles and recommendations.
Page 5, line 209. Is this a new chapter? Maybe you should start with 2.6?
Done.
Page 6, line 235. Is "barrers" intended as "barriers"?
Correct, thank you, amended.
Page 6, line 267. Please, correct with "England".
Done, thank you.
Page 6, line 270. Please, put a comma after Netherlands.
Done.
Page 9, line 385. Please, correct with "higher".
Done.
I hope my comments will help you improve the quality of this manuscript.
Thank you very much for your valuable comments, the manuscript has gained quality through your support.